# Optimization under Uncertainty in the Prize-Collecting Traveling Salesman Problem: An Artificial Intelligence and Simheuristics Approach

C. Tobar-Fernández[*]       A.D. López-Sánchez[†]       J. Sánchez-Oro[‡]

## Abstract

This research addresses the Prize-Collecting Traveling Salesman Problem (PCTSP) under demand uncertainty, a challenge in route optimization where unmet demands incur penalties. Traditional deterministic models fail to capture real-world variability. In this context, a new methodology is proposed that integratess Artificial Intelligence (AI) and simheuristic techniques, which arise from the combination of simulations with heuristics, to improve decision making in uncertain environments. Specifically, Machine Learning models are used to predict demand by obtaining an approximation to the most affine deterministic world assumption, while clustering methods generate realistic demand scenarios. All this giving a more realistic approach replacing the classical ones with Monte Carlo simulations. A simheuristic approach combining GRASP with simulations will be used, aided by Machine Learning methods to improve the evaluation of solutions under stochastic conditions.

## 1   Motivation

The Prize-Collecting Traveling Salesman Problem (PCTSP) has various real-world applications, especially in scenarios involving uncertainty and incomplete information. Such issues may be used to optimize delivery routes, taking into account penalties for missed destinations due to time or capacity constraints, which helps improve the efficiency of distribution systems. The PCTSP models realistic situations as waste collection, healthcare services to infrastructure maintenance among others. Regarding waste collection and management, the PCTSP may model the scheduling of pickups, applying penalties when certain locations cannot be serviced on time, and can also include environmental compliance considerations, Toth and Vigo (2002). If we focus on healthcare service planning, the PCTSP may optimizes routes for healthcare workers, with penalties for unvisited patients, specially when resources like time and personnel are limited. Additionally, in infrastructure maintenance, PCTSP may aid in the planning of site visits for maintenance tasks, where penalties are incurred if all tasks cannot be completed within a single route, particularly in emergency situations where time or resources are constrained. Overall, PCTSP is valuable in environments where uncertainty, time constraints, and resource limitations impact the ability to fully achieve the objectives of the routing problem, Balas (2007).

Optimization under uncertainty is a significant challenge in Operations Research, especially in routing problems like the PCTSP. Traditional approaches rely on deterministic models, which may not adequately capture real-world variability. Demand uncertainty introduces complexity that impacts cost efficiency and service reliability. This research aims to address these challenges by integrating

[*]University Rey Juan Carlos. Email: c.tobar.2024@alumnos.urjc.es
[†]Universidad Pablo de Olavide. Email: adlopsan@upo.es
[‡]University Rey Juan Carlos. Email: jesus.sanchezoro@urjc.es

XVI XVI Congreso Español de Metaheurísticas, Algoritmos Evolutivos y Bioinspirados (maeb 2025).

Artificial Intelligence (AI) techniques and simheuristics to enhance decision-making in uncertain environments.

## 2 Problem Definition

The Traveling Salesman Problem (TSP) is a classical and widely studied problem in combinatorial optimization, with numerous real-world applications ranging from logistics and transportation to manufacturing and bioinformatics. Traditionally, the TSP seeks to determine the shortest route for a salesman to visit a set of cities, ensuring that each city is visited exactly once before returning to the starting point.

This research focuses on a variant of the TSP known as the Prize-Collecting Traveling Salesman Problem (PCTSP) presented by Balas in 1989, Balas (1989), where the objective is not only to minimize travel costs but also to account for penalties incurred from undelivered demands at certain nodes. In the PCTSP, a salesman must balance the costs associated with traveling and the penalties, this introduces an additional layer of complexity. In this work, however, we will associate a cost per unit of distance traveled and a cost per of non-delivered demand unit. In this way we will minimize the total cost by approaching this multi-objective optimization problem on a single objective.

Previous studies have explored exact and heuristic algorithms for PCTS,P Clímaco et al. (2021), including techniques such as integer linear programming (ILP), Souza et al. (2019) and branch-and-bound. These approaches are for deterministic PCTSP, the problem addressed in this study is the extension of PCTSP to include demand uncertainty. The objective is to develop methodologies that can effectively manage uncertain demands while minimizing the total cost.

The primary methods for addressing uncertainty in the PCTSP are simheuristics, Juan et al. (2018), which integrate simulation techniques with heuristic algorithms to adeptly handle stochastic factors, such as unpredictable demand. Simheuristics are hybrid approaches that combine the advantages of simulation and heuristics to deliver effective answers in complex and unpredictable contexts. Simheuristics provide a robust framework for assessing solutions under uncertainty by simulating diverse demand scenarios and employing heuristics to direct the search for viable answers. This methodology seeks to improve solution quality while guaranteeing the resilience and flexibility of decisions amid variable demand, a significant problem in practical applications of the PCTSP.

## 3 Hypothesis

Combining Machine Learning insights with simheuristic approaches, such as GRASP with Monte Carlo simulations, improves solution robustness and quality, particularly in stochastic environments.

The central hypothesis of this research is that integrating Artificial Intelligence techniques along with simheuristic methods, enhances the efficiency and accuracy of solving the PCTSP under demand uncertainty. Specifically utilizing clustering models to generate scenarios yields more realistic and diverse representations of uncertain demands compared to traditional random sampling methods and utilizing predictive models, instead to use expected values, allow for more effective deterministic optimizations by reducing uncertainty.

A potential limitation of this methodology is that since the scenarios are derived from historical data, a sufficiently extensive dataset is required to generate a large set of simulations. If the historical data are limited or not representative of future conditions, the quality and reliability of the simulated scenarios may be compromised.

## 4 Objectives

The objective of this research is to explore innovative approaches for solving the PCTSP under demand uncertainty. In particular, tests will be conducted using a real, enriched dataset of historical demand data to validate the proposed methodologies. The following specific goals have been identified to guide the research process:

- Predict demands using ML models (e.g., Linear Regression, Lasso, Ridge, Random Forest, Neural Network, XGBoost) to approximate the stochastic problem in the deterministic world.
- Develop AI-enhanced scenario generation techniques: Utilize clustering models (e.g., K-nearest neighbors) to generate realistic demand scenarios, capturing complex patterns in historical data.
- Explore simheuristic methods: Integrate metaheuristics like GRASP with simulation techniques to handle uncertainty, enhancing solution robustness and stability.
- Evaluate solution performance under uncertainty: Measure solution quality in the stochastic world, considering expected value, worst-case performance, stability (dispersion), and feasibility (reliability).
- Benchmark against classical approaches: Compare proposed methodologies with traditional stochastic optimization methods to validate effectiveness and efficiency.

# 5  Methodology

When addressing problems under uncertainty, it is possible to employ metaheuristics such as GRASP in combination with simulations. These techniques are known as simheuristics. This project aims to examine the performance of such techniques under uncertainty and to explore the potential of combining these methodologies with Machine Learning models to enhance their outcomes, given the extensive information provided by the data.

One of the key challenges in solving stochastic problems lies in evaluating the quality of a solution. Specifically, it raises the question of how to determine whether a solution is good or bad. To address this, one can assess the solution in two different settings: a hypothetical deterministic world or a stochastic world, Juan et al. (2015).

In the stochastic setting, several parameters can be evaluated depending on the objective. For instance, one may be interested in assessing whether a solution performs well on average or in the worst-case scenario. Additionally, it might be relevant to evaluate the stability of the solution by measuring its dispersion or to ensure its reliability by examining its feasibility rate.

The simheuristic approach to this problem involves several key steps. First, the stochastic problem is approximated by replacing the random demands with their expected values, thus creating a deterministic version of the PCTSP. This approximation provides a simplified version of the problem, which serves as the basis for the next step.

Following this approximation, a metaheuristic is employed to find an initial solution for the deterministic version of the PCTSP. This initial solution is then subjected to a simulation phase in which multiple demand scenarios are generated and evaluated. In this phase, demand values for each customer are generated based on their probability distributions using Monte Carlo simulations with a lognormal distribution. Subsequently, the performance of the initial solution is assessed across all simulated scenarios, allowing for a more comprehensive evaluation of its effectiveness under uncertainty.

This project proposes two enhancements to this traditional simheuristic approach. First, Machine Learning models will be integrated to improve the initial deterministic approximation. By leveraging the predictive power of Machine Learning, it is anticipated that the initial solution will achieve a better fit compared to the conventional approach of using expected values alone.

Second, scenario generation will be enhanced through the use of clustering methods, such as K-Nearest Neighbors (KNN), to produce more realistic scenarios. This method is expected to provide a more accurate representation of demand variability, leading to more robust solutions.

Moreover, evaluating complex stochastic metrics, such as feasibility or dispersion, can be computationally expensive when incorporated throughout the metaheuristic model. To address this issue, the simulation phase will be divided into two stages. In the first stage, a rapid simulation will be conducted to assess the expected performance of the solution. After this preliminary evaluation, the top five to ten solutions will be selected and analyzed using a more detailed and computationally intensive simulator. This two-step approach will enable a more precise assessment of the most complex stochastic metrics, ensuring a rigorous evaluation of the solutions' effectiveness under

uncertainty. To avoid artificial aggregation of criteria, Pareto-front analysis and alternative weighting strategies will be explored.

The proposed methodology, due to the combination of GRASP, Monte Carlo and multistage simulations, has a high computational cost. To mitigate the computational challenges, techniques such as parallelized GRASP can be employed, along with convergence-based stopping criteria to optimize runtime.

This methodological framework is expected to provide valuable insights into the behavior of simheuristic techniques under uncertainty, while also demonstrating the benefits of integrating Machine Learning models to enhance decision-making processes in stochastic optimization problems.

Although real-world validation with a specific company is not always feasible in preliminary studies, tests will be performed with reduced real instances, as well as with larger artificial instances that provide a solid basis for evaluating the effectiveness of the methodology.

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
