# OpenReview forum: "Optimization under Uncertainty in the Prize-Collecting Traveling Salesman Problem: An Artificial Intelligence and Simheuristics Approach"
_MAEB/2025/Projects_Track — MAEB 2025 Proyectos_

### Official Review · Reviewer_rEFj · 2025-03-17
**Optimization under Uncertainty in the Prize-Collecting Traveling Salesman Problem: An Artificial Intelligence and Simheuristics Approach**

**Rating:** 3
**Confidence:** 2

**Review:**

According to the abstract, “This project addresses the Prize-Collecting Traveling Salesman Problem (PCTSP) under demand uncertainty”.

Unfortunately, I do not have a clear idea of the main goal of the project. Artificial Intelligence techniques, simheuristics, and other techniques are mentioned but, looking at the hypothesis and the objectives: is the goal to create a “generator”, that is, a tool that can be used to create scenarios with different characteristics? The tool would learn ML models from real instances (first objective)?

How does it link with the sentence “In this way we will minimize the total cost by approaching this multi-objective optimization problem on a single objective”: The problems that will be created are constrained, not real-world instances? If this is the case, is the generator a useful tool (in case a generator is what authors propose)?

In addition, is PCTSP a real problem nowadays? Can the authors include some references about this topic?

I would appreciate it if the authors could revise and rewrite the text to get a clearer idea of the project, the main objective, and its phases.

Minor comments:

Please check references and their format (numbers and text).

There also some typos in the text. Please revise the whole document.

---

### Official Review · Reviewer_iCcA · 2025-03-17
**Review for the project "Optimization under Uncertainty in the PCTSP"**

**Rating:** 5
**Confidence:** 4

**Review:**

The project proposal seems interesting and well-focused. It proposes the integration of artificial intelligence (AI) and simheuristics to address the Prize-Collecting Traveling Salesman Problem (PCTSP) under demand uncertainty. Overall, it is a well-structured proposal with a correct methodology. Although not a completely disruptive proposal, it can be a useful contribution if it is shown that the proposed improvements add a tangible benefit.

The innovation in the combination of AI and simheuristics could be significant, since the use of Machine Learning models to predict demand and clustering techniques to generate realistic scenarios provides a more robust approach than simple traditional random sampling. An initial deterministic approximation strategy followed by detailed simulation in several stages can help find more robust and reliable solutions. The intention to test the methodology with an enriched dataset gives it a very valuable empirical component. And considering metrics such as the mathematical expectation, the worst case, stability and reliability will allow a detailed view of the quality of the solution.

Regarding the innovation represented by this proposal, it is worth saying that the use of simheuristics in stochastic routing problems is not new. Several studies have applied Monte Carlo combined with heuristics/metaheristics for similar problems, therefore, the differential value of the proposal should better justify how it surpasses these previous studies. On the other hand, the use of Machine Learning to predict demand and improve simulation is interesting, but the proposed models (Linear Regression, Lasso, Ridge) are quite basic and I have doubts about whether these models really add value with respect to more classic approximations (such as historical averages or parametric distributions). These models may be sufficient to approximate demand, but it may be necessary to explore the possibility of using neural networks or models based on time series (such as ARIMA or LSTMs). If ML is only used to make a demand prediction that is then added to an existing model, then the innovation is marginal. Regarding clustering, it must be said that the use of KNN to generate realistic scenarios is an interesting idea, but it is not clear whether it can really represent an improvement over traditional techniques (e.g. pure Monte Carlo or empirical distributions). Other approaches such as Gaussian Mixture Models (GMM) or Bayesian Networks could give more flexibility to the simulation scenario.

I find that the proposal has certain methodological limitations. First of all, due to the high computational cost that it will surely have, since the combination of GRASP, Monte Carlo and multi-stage simulations can be computationally expensive. It will probably be necessary to study how to reduce the calculation time without losing quality. The experiments need a justification because it does not mention what data volumes and what execution times are expected. Secondly, regarding the quality assessment of the solutions, the article mentions different metrics (expectation, worst case, stability, etc.), but it is not clear how they will be weighted. Multi-objective optimization techniques, such as Pareto-fronts, could be explored to avoid forcing an artificial aggregation of criteria. The use of historical data is mentioned, where it is very possible that this approach will work, but it is not indicated whether the methodology will be tested in a real case with a specific company or sector. It would also be interesting to know what happens in scenarios with very scarce data or with very irregular demand distributions.

In conclusion, the idea of ​​combining AI, heuristics and simulation to improve decisions in an uncertain environment is solid and can be a useful contribution. The innovation may be less than it seems if it only slightly improves existing approaches. The computational cost could make it not very applicable in real environments. A clear validation with practical cases to demonstrate its real utility would be needed.

---

### Decision · Program_Chairs · 2025-03-20

Accept